

# Role of moisture transport for Central American precipitation

Ana María Durán-Quesada[1,2], Luis Gimeno[3], Jorge Amador [1,2]

[1]Department of Atmospheric, Oceanic and Planetary Physics, University of Costa Rica, Costa Rica.
[2]Center for Geophysical Research, University of Costa Rica, Costa Rica.
[3]Environmental Physics Laboratory, University of Vigo, Spain.

*Correspondence to*: Ana María Durán-Quesada (ana.duranquesada@ucr.ac.cr)

**Abstract.** A climatology of moisture sources linked with Central American precipitation was computed based upon
Lagrangian trajectories for the analysis period 1980-2013. The response of the annual cycle of precipitation in terms of
moisture supply from the sources was analysed.  Regional precipitation patterns are mostly driven by moisture transport
from the Caribbean Sea (CS). Moisture supply from the Eastern Tropical Pacific (ETPac) and Northern South America
(NSA) exhibits a strong seasonal pattern but weaker compared to CS. The regional distribution of rainfall is largely
influenced by a local signal associated with surface fluxes during the first part of the rainy season, whereas large scale
dynamics forces rainfall during the second part of the rainy season. The Caribbean Low Level Jet (CLLJ) and the Chocó Jet
(CJ) are  the main conveyors of regional moisture, being key to define the seasonality of large scale forced rainfall.
Therefore, interannual variability of rainfall is highly dependent of the regional LLJs to the atmospheric variability modes.
The El Niño-Southern Oscillation (ENSO) was found to be the dominant mode affecting moisture supply for Central
American precipitation via the modulation of regional phenomena. Evaporative sources show opposite anomaly patterns
during warm and cold ENSO phases, as a result of the strengthening and weakening, respectively, of the CLLJ during the
summer months.  Trends in both moisture supply and precipitation over the last three decades were computed, results suggest
that precipitation trends are not homogeneous for Central America. Trends in moisture supply from the sources identified
show a marked north-south seesaw, with an increasing supply from the Caribbean Sea to northern Central America. Long
term trends in moisture supply are larger for the transition months (March and October). This might have important
implications given that any changes in the conditions seen during the transition to the rainy season may induce stronger
precipitation trends.

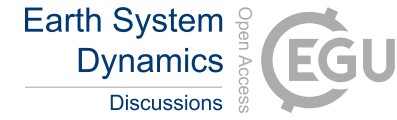

## 1 Introduction

Central America consists of a relatively thin strip of land surrounded by three large warm-water bodies, namely the Caribbean Sea, the Easternmost tropical Pacific (ETPac), and the Gulf of Mexico (GoM). The region is known to be highly vulnerable to droughts, floods and associated landslides. A high percentage of the natural disasters reported in the area are

linked to hydrometeorological events (*Alfaro et al., 2014*), which produce significant economic losses, as well as human casualties. Socio-economic development in the region is somewhat constrained by the annual cycle of rainfall, given that agriculture is one of the main economic drivers. Any improvement in terms of regional weather forecasting remains elusive as long as knowledge of regional precipitation and its sources is limited. The overall effect of major tropical disturbances on regional patters of precipitation lacks scientific understanding, and this too affects the accuracy of forecasting. The role of

the Intra Americas Sea (IAS) in the relevant tropic-midlatitude interaction and feedback processes also requires further investigation (*Douville et al., 2011*). In very broad terms, the regional weather and climate are influenced by large sources of latent heat, strong easterly winds, the North Atlantic Subtropical High (NASH), high sea surface temperatures (SSTs), and intense precipitation (*Wang, 2007*). Regional terrain, including topography and diverse vegetation, found to be of importance (*Lachniet & Patterson, 2007*) are still not properly represented in numerical . The horizontal 'seesaw' observed in Central

American precipitation is often attributed to the effect of the continental divide. Drier conditions in northern Central America and the Pacific slope are in contrast with a wetter Caribbean side. The annual cycle of precipitation is characterised by a bimodal distribution that exhibits a minimum in July-August and maximum values in June and September-October. This bimodal distribution of precipitation, known as Mid-Summer Drought (MSD, after *Magaña et al., 1999*), is more marked on the Pacific side. On the Caribbean slope, heavy precipitation is observed during May and June, followed by a drastic

reduction in late June, leading to a drier and less cloudy July-August (*Taylor & Alfaro, 2005*). Several authors have pointed to the seasonal migration of the Inter-Tropical Convergence Zone (ITCZ), deep convection, low level moisture transport, cyclone activity, and mid-latitude air intrusions as being the main drivers of regional precipitation (*Schulz et al., 1997, Amador et al., 2006, Durán-Quesada et al., 2010 , among others*). Studies on the effect of large scale structures such as the ITCZ on regional precipitation are scarce (*Hidalgo et al., 2015*), and some important interactions, including links with the

tropical Pacific, are not fully understood. The region is influenced by deep convection and highly active stratified precipitation. A large deep convective core is located over the Panama Bight (*Zuluaga and Houze, 2015*), with an extended area of stratiform precipitation that becomes relevant for overall rainfall. The effect of mid-latitude interactions is known to be mostly related to the occurrence of rainfall during the dry period on the Pacific slope between November and February (*Zárate, 2013; Sáenz and Durán-Quesada, 2015*).


Previous works by the authors using global (*Durán-Quesada et al (2010))* and limited domain (*Durán-Quesada (2012))* *trajectories* identified the sources of moisture linked to Central American precipitation. Their results highlighted the role of the Caribbean as the main source of moisture and the relevance of the Caribbean Low Level Jet (*Amador 1998 ; 2008*) as a



regional moisture conveyor. *Wang et al (2013)* analysed the moisture transport from the Caribbean to the Pacific across the
Americas, highlighting the CLLJ as an interbasin moisture transport mechanism. These results, together with those of several
authors including *Xu et al (2005), Leduc et al (2007)*, and *Richter & Xie (2010)* show coherence on the relevance of moisture
transport from the Caribbean Sea. However, other regional moisture suppliers are poorly understood and the key processes
that drive Central American precipitation remain unclear. The importance of other structures including the Chocó Jet (CJ)
have mainly been considered in terms of interannual variability modes (*Poveda & Mesa, 2000*) and Mesoscale Convective
Systems (MCS, *Zuluaga & Poveda, 2004*). The response of regional precipitation to moisture supply from evaporative
sources at different timescales has not yet been fully addressed. The connectivity between moisture transport from northern
South America (NSA) to Central America also requires further consideration. Moreover, long-term trends in moisture supply
and their effect on detected precipitation and low level wind trends have never been documented for the Central American
case.
This study is devoted to the long term analysis of the importance of moisture supply to Central American precipitation, from
the annual cycle to some aspects of interannual variability. Moisture transport controlled by regional LLJs is analysed to
better explain the transport of moisture at intraseasonal scales and to study the modulation of these scales by interannual
modes of atmospheric variability (i.e., ENSO). Trends in significant precipitation and moisture supply are analysed in order
to assess consistency among precipitation trends and detected tendencies in terms of moisture supply . The remainder of the
paper is organized as follows, Section 2 provides information on the methods and data used. The analysis of moisture supply
for Central American precipitation from the identified sources is presented from a climatological perspective in Section 3.
The annual cycle of moisture supply to Central American precipitation is then discussed further in Section 4. Section 5
emphasises the response of precipitation to moisture supply under ENSO conditions. An analysis of moisture supply and
long term trends in precipitation is given in Section 6, and some concluding remarks are presented in Section 7.


## 2 Data and Methodology

In the present study the identification of moisture sources is based on the source-receptor relationship of the hydrological
cycle. A Lagrangian numerical water vapour tracer approach was implemented. Considering that the region is characterised
by intense rainfall and high rates of evaporation, we used the method developed by *Stohl & James (2004)*, because unlike
other methods it provides a diagnostic of the net freshwater flux (E-P) rather than of evaporation (E) alone, (Gimeno et al.,
2012). Global Lagrangian backward trajectories were generated using the Lagrangian particle dispersion model FLEXPART
version 8 (*Eckhardt et al., 2008)* initialised with ERA-Interim Reanalysis data *(Dee et al., 2011)* using a 0.75-degree
horizontal resolution with water vapour as a tracer. Input data consisted of analyses every 6 hours (0000, 0600, 1200 and
1800) with three-hourly forecasts for intermediate times (0300, 0900, 1500, 2100). Ten-day Lagrangian backward
trajectories were generated in a global domain for a total of $2.5 \times 10^6$ particles uniformly distributed within it, with each





receiving the same mass. The accuracy of the method used to detect precipitation was assessed for Costa Rica by comparing rainy days detected from the trajectories with rainy days based on TRMM (Tropical Rainfall Measuring Mission, *Huffman et al., 2007*) and raing gauges located across the country. The trajectories method captured up to 86 % (82%) of rainy days for the analysis domain detected by TRMM (rain gauges) over the period 2007-2012. Notice that for the Central Valley area, the

matching of rainy days between the used rain gauges and TRMM is on average 90%. Accuracy of TRMM rainfall amount estimates for Central America deserves validation, however this is not the focus of the present work. Following the approach of *Stohl & James (2004)*, precipitating air masses over the Central American region were tracked backwards in time for 10 days. The net fresh water flux was estimated for daily aggregates, with daily information averaged on a monthly basis to obtain the 1980-2013 climatology. Moisture exports from the identified evaporative sources to Central America were

quantified to generate long-term time series of moisture export.

A set of indices was used to evaluate the relationship between the contribution of the moisture sources to Central American precipitation and regional mechanisms. To quantify the intensity of the regional low-level jets, a CLLJ index was computed as the 925-hPa zonal wind averaged in the region 12.5-17.5N, 80-70W, based on previous work by Amador (1998), Amador et al. (2006), and Amador (2008), and similar to that defined by *Wang (2007)*. A CJ index was calculated as the 925-hPa

zonal wind averaged for the region of 5S-7.5N, 80W, following the definition of *Poveda & Mesa (2000)*. Outgoing longwave radiation (OLR) from *Chelliah & Arkin (1992)* was averaged for 5 regions to represent an estimate of convection over North, Northwest, Central, Southwest, and Southeast Amazon following a similar selection of area to *Marengo et al (2001)*. High resolution precipitation data for Central America were obtained from the Climate Hazard Group InfraRed Precipitation with station data archive (CHIRPS, *Funk et al., 2015*). Additional information used includes evapotranspiration

from MODIS *(*Moderate Resolution Imaging Spectroradiometer, *Mu et al., 2007)* for 2000-2010 and the Multivariate ENSO index (MEI, *Wolter & Timlin, 1998*).

## 3 Results and discussion

### 3.1 Moisture supply and rainfall: behavior and relevance of Central American precipitation moisture sources.

*Durán-Quesada et al (2010)* diagnosed the moisture sources for Central American precipitation based on 2000-2004 ERA-40 and FLEXPART computed trajectories, reporting the Caribbean Sea as the main moisture source with a secondary contribution from the southern portion of the ETPac. Due to the short time span analysed, the authors of this study were unable to reveal any further information either on the climatology of the evaporative moisture sources linked to regional precipitation or on regional moisture transport and its variability.


Despite the Caribbean Sea (CS) being a continuous supply of moisture, moisture transport presents seasonality featured by the horizontal extent of the source. The CS extends from the Central American east coast to the southeastern Caribbean with





a consistently strong intensity in terms of moisture supply between December and May along the Central American eastern coast (see Fig. 1.a-b). The maximum intensity of the CS source shifts to the east as the Caribbean SST increases (see Fig. 1.c). Moisture exports from the CS reach up to 10 mm/day except during September-October-November (SON) when the maximum supply decreases to c.5 mm/day. . It is notable that the diminishing of the CS as a source during SON is accompanied by a reduction in the easterly flow (see black vectors in Fig. 1.c). During these months, the Western Hemisphere Warm Pool (WHWP, Wang and Enfield, 2001) develops its Pacific component, the Caribbean SST falls, and moisture is largely dragged by the cyclonic systems that develop in the Atlantic-Caribbean. The arrival of moisture exports from the CS to Central America is limited to long-range transport and by a reduction in moisture availability.  Moisture transport from the GoM to the region becomes noticeable from late September to February, even though the contribution is much smaller than that from the CS. As The relevance of the GoM as a moisture source for the region is constrained by mid-latitude interactions. This source acquires importance in association with winter circulation patterns related to cold surges, which have been found to contribute to the precipitation over Central America during a relatively dry period (*Sáenz and Durán-Quesada, 2015*).

Moisture supply from the ETPac starts to develop in late May and reaches a maximum intensity between August and September (Fig. 1.c).  The ETPac source is located in a region characterised by strong evaporation (*Amador et al., 2006*). Furthermore, the peak in its annual cycle is coherent with the maximum intensity of the CJ (*Poveda & Mesa, 2000*). As the easterly flow decreases and the CJ intensifies, moisture supply from the ETPac is enhanced (Fig. 1.d). This behaviour highlights the interplay between the CLLJ and the CJ in regional moisture transport. In addition, the intensity of the deep convection in the ETPac provides a suitable environment for the release of latent heat and enhanced surface evaporation. The latter may lead to an increase in moisture availability related to the ETPac source. Moreover, contributions from the ETPac source are also forced by large-scale processes including ITCZ movements, shallow meridional circulation in the region (Zhang et al. 2004), and the development of deep convection.

The identification of moisture sources suggests that northern South America represents a remote supply of moisture for Central American precipitation. The Orinoco river basin is found to be an evaporative source with an annual cycle similar to that of the CS. The Magdalena river basin is identified to play an important role in long-range continental moisture exports to Central America. In this case, the moisture supply is stronger in June-July-August (JJA). It is worth noting that the Magdalena and Orinoco river basins are also the principal moisture sinks in northern South America (*Meade 2007* ; *Poveda et al. 2001* ). Despite this, they provide a summer-time contribution to Central American rainfall. the transport of moisture from these remote continental sources is constrained by regional convective activity and local precipitation.

Central America is itself an evaporative moisture source, as suggested by the large positive values of (E-P)[10]. The annual cycle is well defined and northern Central America is the primary destination of the source, the transport being stronger from December to early June. It is beyond the scope of this work to quantify the evapotranspiration; however, it is important to





assess the consistency between this variable and the activity of the local moisture source. By comparing Figs. 1 and 2, it can be seen that the strong seasonal signal of evapotranspiration from *MODIS* is coherent with the seasonal characteristics of the identified supply of moisture from Central America. A maximum evapotranspiration for the area of Patuca National Park,

Bosawás Natural Preserve, and Río Plátano Biological Preserve in Honduras is exhibited for March-April-May (MAM). Other areas with significant evapotranspiration are El Mirador National Park, Sierra Del Lacandón National Park, and Biosfera Maya Biological Preserve in Guatemala, as well as Calakmul Biosphere Preserve and Montes Azules National Park in Mexico . This finding is in good agreement with the highest positive value of $(E-P)^{-10}$ detected over northern Nicaragua, Honduras and the Yucatán Peninsula. Because evapotranspiration depends on plants (*Brummer et al., 2013*), the highest

values tend to be found in regions with dense vegetation. Fig. 1.b indicates that the region becomes a strong moisture supplier during the first rainy season (MAM) suggesting the regional importance of surface-precipitation feedback mechanisms. Smaller moisture contributions from the Pacific slope of Central America are consistent with the known horizontal precipitation 'seesaw' between the Caribbean and Pacific sides of Central America. However, a more fundamental analysis is required to establish the link between local moisture supply and evapotranspiration, as well as to quantify

moisture recycling and its direct relationship with rainfall.

### 3.2 Annual cycle of moisture transport to Central American precipitation and the role of the LLJs

Moisture transport from each identified evaporative moisture source was computed from the trajectories dataset. The annual cycle of moisture transport from the CS, as shown in Fig. 3.a, suggests a relatively constant moisture supply to Central

America that decreases slightly between September and November, which is a period of intense development of tropical cyclones over the Atlantic basin. More detailed analysis allows us to determine that the overall moisture supply to Central America from the CS shows strong variations across Central America. In fact, computing the moisture transport for the Central American countries reveals significant differences in the annual cycle of moisture transport. Moisture transport from the CS to Belize and El Salvador is small (0.2 Sv), and nearly constant throughout the year, due in part to a marginal

exposure to the moist air flow. Transport to Honduras, Guatemala and Nicaragua is consistent with the overall annual cycle depicted in Fig. 3.a. The main difference is that moisture transport from this source is larger for Nicaragua (up to 1.5 Sv) and is more intense from October to May compared with transport to Honduras and Guatemala. Transport from the Caribbean Sea to southernmost Central America shows a characteristic annual cycle characterised by a more intense moisture supply in December-January-February (DJF). It is worth noting that in summer the transport to Costa Rica and Panama decreases,

while increasing for Honduras, Guatemala and Nicaragua at this time. Country estimates of CS moisture transport reflect the strong dependence it has on the position and intensity of the CLLJ.

Moisture supply from the ETPac to Central American precipitation is markedly bimodal, as shown in Fig. 3.b. Even when this source is an order of magnitude smaller than the Caribbean supply, moisture transport from the ETPac still peaks during

the months that characterise the transition to the rainy season. The July minimum is coherent with the intensification





(decrease) of the easterly (westerly) flow and partly coincides with the presence of the MSD over the Pacific slope of Central America. Unlike the transport from the Caribbean, the bimodal moisture supply pattern is observed for the whole Central American region. Transport is greater to Panama, Nicaragua and Costa Rica, with the two last countries showing the strongest July minimum. The secondary peak of transport to Panama represents up to double the transport that occurs during

May (not shown). The fingerprint of the transport of moisture from the GoM is also well defined, with transport increasing during October and continuously decreasing until it falls to a minimum after May (Fig. 3.c). It can be seen that the interannual variability is relatively small and that extreme values shown as outliers (marks) are significant, showing much stronger transport. The fact that moisture supply from the GoM peaks during a period seeing the entry of mid-latitude systems is highlighted, because this provides information relevant to the links between mid-latitude interaction and rainfall

during the dry season. It is also worth mentioning that even when moisture supply is mostly constrained to the northern portion of Central America, Nicaragua receives approximately 0.10 Sv when the source is active.

Local contributions are in broad coherence with the first part of the rainy season in the region (Fig 3.d). In agreement with the seasonal distribution of evapotranspiration according to MODIS estimates (not shown), Guatemala, Honduras and

Nicaragua receive their largest supply from inland evaporative sources, suggesting that moisture recycling could follow a similar annual cycle. This latter view may support the hypothesis of a difference between the first and second parts of the rainy season, which suggests that the second part is driven by large-scale dynamics rather than by the local processes that drive the first part of the rainy season. A rigorous quantification of moisture recycling in the region is key to assessing the role of local contributions to precipitation. This is not a simple matter because vegetation dynamics and temperature are both

fundamental parts of the problem, and intensive changes in land use are well known in the region (*Aide et al., 2013*).

Transport from the remote evaporative source from the Orinoco and Magdalena river basins peaks in the summer months. It shows a moderate interannual variability and a strong influence of extremes (Fig. 3.e). A more detailed country-level analysis reveals a slightly different latitudinal supply regime from this source. While a bimodal annual cycle of moisture supply is

observed for Nicaragua and northern countries with maxima in June and September, a single summer peak is seen in the moisture contributions to Costa Rica and Panama. Furthermore, the moisture supply to Panama (max 0.41 Sv) is around twice as high as the transport to Costa Rica (0.19 Sv) and more intense than for the other countries. Supply from this remote source is relevant because it may help us to understand how northern South American processes are connected with Central American rainfall distribution and are further linked with the Caribbean Sea dynamics. Moisture transport from this source is

constrained by the availability of moisture, which implies that the supply is dependent on precipitation over the basin and on high evaporation rates. The marked seasonality of the moisture transport from this source is in good agreement with the known annual cycle of moisture sinks in northern South America. In the case of the Orinoco basin, the precipitation follows a similar annual cycle to the moisture supply from Northern South America . Precipitation from the Musinacio station (located by the Mapire river mouth), for example, exhibits a marked peak during June (see Fig. 2 in *Dezzeo et al., 2002*). At





the same time, the Magdalena Valley  experiences a relative minimum of precipitation, suggesting that in summer evaporation exceeds precipitation in the region, enhancing the atmospheric moisture. The moist air masses move to the northwest as a result of the strengthening of the easterly flow.

Regression of the evaporative sources against the CLLJ and CJ indices (Figs. 4.a and 4.b) show the contrasting response of
the two main oceanic sources to these LLJs. As the easterly flow increases, the development of the CLLJ favours transport from the CS, while transport from the ETPac is largely inhibited (Fig 4.a). The opposite is observed for the CJ (Fig. 4.b). The strong link between the arrival of moisture from the oceanic sources and the LLJs shows the importance of these structures to the regional distribution of precipitation. It is worth noting that the CLLJ not only accounts for the dragging of moisture from the Caribbean but is also able to modulate moisture transport from the tropical Atlantic and northern South America
into the mid-latitudes of both hemispheres. Here again, moisture contributions from northern South America depend on both regional and large-scale processes. On the one hand, the interplay between the Orinoco and Magdalena river basins as sinks for Amazon moisture implies a moisture availability for Central American precipitation constrained by Amazon convection. Indeed, moisture supply from northern South America increases with Amazon convection, as suggested by the minimum OLR values for selected regions over the Amazon (not shown). Convection over the Amazon enhances evapotranspiration
processes in the region and provides moisture to the Magdalena and Orinoco basins. Moisture held by the atmosphere is available to be transported and in the presence of an efficient conveyor (e.g., the CLLJ) it is then transported to Central America. The intensification of moisture transport from this source between May and September is in agreement with the reduction in rainfall for several locations in northern Colombia (*Poveda et al., 2005*).

Moisture supply from the ETPac relates to a low CLLJ intensity but also to the increase in the extent of the Pacific
component of the WHWP. Large-scale convection also plays a role in the modulation of the ETPac moisture transport, which is inhibited by deep convection such as that related to the ITCZ. A large negative correlation between the OLR averaged for the easternmost part of the ITCZ and the ETPac moisture source was found to be most pronounced in summer-autumn (not shown). The identified mechanisms are largely related to the development of precipitation in the second part of the rainy season. The moisture supply that may enhance the first part of the rainy season in the region is not fully understood. The
strong link between the distribution of precipitation and the activity of an evaporative source located over Central America suggests a relationship with surface conditions. It has been mentioned that evapotranspiration may play a key role in supplying moisture during the first part of the rainy season. This connection relates to the issue of vegetation-precipitation controls. Therefore, the quantification of evapotranspiration and some knowledge of the vegetation fraction would give new insights and scientific knowledge of the processes that dominate the first part of the rainy season at different timescales.


The interpretation of the relationship between moisture supply and rainfall over a defined region is often an issue. In this study, we aim to provide information about how important is the moisture supply for regional rainfall. Once the moisture sources were characterized, the moisture supply from each source was estimated using a Lagrangian approach. The



correlation between time series of moisture supply and spatial CHIRPS rainfall estimates for the detected rainy days (90%
significance) was computed, results are summarized in figure 5. Moisture transport from the CS favors rainfall in northern
Central America Caribbean slope early in the year (Fig. 5.a). It becomes clear from Fig. 5.b that moisture transport has little
to do with the first rainy season, providing evidence on the consideration that this rainy period is mostly locally forced. See
also Fig. 5.i, from which it can be observed that overall moisture supply from the continental region largely decrease during
the first rainy season as very little moisture is available. For mid summer, CS transport (Fig. 5.c) shows a remarkable result,
which is, moisture transport while the CLLJ reaches its maximum contributes to Caribbean Costa Rica rainfall increase.
Moreover, as the transport increases, Central American Pacific coas becomes drier. The area featured by negative correlation
values (blue in Fig 5.c) is clearly identified as the Central American Dry Corridor (CADC). Here we highlight that fact that
the intensification of moisture transport from the CS and associated mechanisms play a major role in the establishment of
the CADC. In contrast, as the CLLJ retreats in 5.d) autumn and the CJ is intensified, rainfall increases aided by the moisture
supply from the now active ETPac source (Fig.5). The modulation of the Pacific rainfall annual cycle, therefore the depth of
MSD results from the interplay between the regional LLJs and of course its associated dynamics. The effect of the GoM
moisture supply is small, still, its influence might contribute to reduce the impact of the MSD period rain deficit (Fig.5. e).
Transport from Norther South America iaccounts as mentioned before, mainly for sourthern Central America (Fig. 5.g).
However, its relevance during summer time also hints to role of its moisture conveyor , the CLLJ, to be key for the
distribution that features the CADC and is shown by the drying linked to the well known MSD (Fig. 5.h).

### 3.3 Response of oceanic moisture sources to ENSO forcing

The integrated positive net freshwater fluxes were composited for warm and cold ENSO events at a monthly time scale
based on a +/- 0.75 ºC SST anomaly using the MEI. The base line period for the computation of the anomalies was 1981-
2010. The results reveal a large deviation in the position and intensity of the evaporative moisture sources compared to
neutral years. Figure 6 shows the warm-cold ENSO differences for positive (E-P)$^{-10}$ values, suggesting a large variability in
the supply from the oceanic moisture sources. During boreal winter, moisture supply from the Caribbean decreases while
inland and coastal Pacific (E-P)$^{-10}$ values increase slightly (Fig. 6.a). The subsequent reduction in Caribbean moisture
transport is attributed to the response of the deceleration of the CLLJ under warm ENSO conditions in winter time (*Amador
et al. 2006, Amador 2008, Maldonado et al., 2015*). Stronger differences are detected as the first rainy season starts, with a
greatly reduced contribution from the southwest Caribbean as shown by the dark blue shading in Fig. 6.b. In contrast,
moisture supply from the northern Caribbean is enhanced for warm ENSO conditions, similar to the evaporative source
observed over Nicaragua and Honduras (red shades in Fig. 6.b). Under cold ENSO conditions, moisture supply from the
Caribbean results in a dipole, from which the inhibition of moisture transport from the southern Caribbean becomes evident.
A small reduction in the positive (E-P)$^{-10}$ values over the northern portion of the Orinoco river basin is also noted. In contrast,
as the CLLJ strengthens and the warm ENSO causes the easterlies to accelerate, the Caribbean Sea moisture source is





reinforced. The red shading in Fig. 6.c over the Caribbean (and northern Colombia) suggests that a warm ENSO favours the intensification of moisture supply from this source, aided by the strengthening of the CLLJ. The increase of moisture

transport from this source under El Niño conditions is consistent with the known surplus of precipitation on the Caribbean slope of Central America for a warm ENSO (Fig. 11 of *Amador 2008*). The composite differences also show good agreement with the proposed moisture conveyor mechanisms as the ETPac supply is reduced for warm ENSO in coherence with the weakening of the south-westerlies over the Pacific.

As a complimentary analysis, the moisture supply from the evaporative sources was computed and the correlation between

the moisture transport and the MEI was estimated. Results for the Caribbean and ETPac sources are shown in Fig. 7. The strongest response to ENSO is observed for moisture contributions from the Caribbean Sea (Fig. 7.a) with a contrasting correlation pattern compared to the response of the ETPac moisture supply (Fig. 7.b). In general, Central America experiences an increase (decrease) of moisture contributions from the Caribbean for warm (cold) ENSO during the transition to the first part of the rainy season. The summer response to ENSO is homogeneous for the Central American region. In

contrast, the winter and spring response is heterogeneous, and the regional response becomes more consistent during the transition to the first part of the rainy season. For this period, the moisture supply exhibits a negative correlation, suggesting that the warm ENSO is linked with a reduction in moisture transport derived from a weakened moisture source in the central Caribbean (Fig. 6.b). This finding is in good agreement with *Maldonado & Alfaro* (2012) who highlighted the intensification of dry conditions under warm ENSO. It is clear that the response of the moisture supply to ENSO is largely influenced by its

effect on the CLLJ and the mean easterly flow, as shown by their identical ENSO response. However, it is important to note that moisture supply from the Caribbean is not driven by transport alone. In particular, under ENSO conditions energy fluxes become important over the Caribbean as a result of SST anomalies and advection of moisture to the atmosphere. *Chikamoto & Tanimoto (2005)* showed that specific humidity differences between the air and sea interfaces are related to asymmetric latent heat flux anomalies linked to ENSO activity. These authors also found that near-surface specific humidity anomalies

dominate the difference in the air-sea humidity anomaly. Because the boundary layer humidity gradient depends on effective evaporation, it can be argued that both the boundary layer depth and the humidity are fundamental to the role of the Caribbean Sea as an evaporative moisture source (Durán-Quesada 2016, personnal communication). As a result of its intensification the CLLJ becomes not only a moisture conveyor structure but also a mechanism able to enhance evaporation by drag, increase the humidity gradient, and also generate surface cooling *(Amador 2008)*. The CLLJ response to ENSO

therefore plays an even bigger role in the modulation of the moisture supply from the Caribbean. Its intensification under warm ENSO increases the moisture transport but also contributes to the increase of atmospheric moisture availability and atmospheric instability, especially in the central Caribbean.

From Fig. 7.a, it is observed that a warm ENSO favours the intensification of a moisture source to the west of the Central American coast during the winter months. Compared with the climatological moisture supply from the ETPac, this winter

contribution intensified by warm ENSO conditions has a different origin. The positive response of the ETPac moisture supply to a warm ENSO during winter (Fig. 7.b) is not completely clear, however. A southward displacement of a weaker



winter ITCZ, caused by changes in the tropical energy flux (*Schneider et al., 2014*) facilitates moisture transport by enhanced westerlies over the Pacific for warm ENSO. However, the location of the moisture source suggests that the origin of this moisture supply is not fully driven by the motion of the ITCZ. The increase in the contributions for warm ENSO is

proposed to be linked to the effect of rain-producing systems. These systems can be associated with the penetration of cold surges, known to increase during El Niño events (Magaña and Vásquez, 2003 ) and to modify the regional distribution of precipitation (*Sáenz and Durán-Quesada, 2015*). During summer, the negative correlation between the moisture supply from the ETPac and the ENSO can be explained by an enhanced deep convection linked to the ITCZ for cold ENSO events as revealed by OLR composites (not shown).


### 3.4 Trends in moisture supply from evaporative sources

Trends in moisture supply from evaporative sources


Central America is often referred to as a climate change hotspot, based on climate projections (*Giorgi 2006, Neelin et al. 2006, Rauscher et al. 2008, 2011*, Diffenbaugh & Giogi, 2012 ; Imbach et al., 2015 ; Nakaewaga et al., 2014 ; Hidalgo et al., 2013). While temperature trends are consistent across many studies, precipitation trends have failed to show any sound significance. An example is given in the observational study of *Aguilar et al (2005)*, which shows a marked significant

temperature trend and highlights the non-significance of any precipitation trends. Understanding trends in the hydrological cycle for the region is an important milestone in providing a solid basis for water management policies in the context of climate change and societal impacts in a region affected by the intensive exploitation of surface water reservoirs (*Arias & Calvo-Alvarado, 2012*). Considering the sensitivity of the region to extreme hydrometeorological events, it is imperative to identify any consistent significant trends towards drier or wetter conditions, in order to support water resources management,

and mitigation and adaptation policies. Long-term trends were detected for moisture supply to Central American precipitation using a modified Mann-Kendall test. The evaluation of these trends was carried out for the countries of interest in order to check for consistency with reported differences in north-south trends.

The strongest trends were detected for oceanic evaporative sources of moisture. A strong seasonal difference was identified

for the Caribbean Sea source (Fig. 8.a), while a consistent trend was reported for the ETPac source. Moisture supply from the CS was found to increase (decrease) by up to $10\times10^{-3}$ Sv/year during March (October). An increasing trend in moisture supply was found during transition months, with values of $1\times10^{-3}$ Sv/year for southern Central America, and $8\times10^{-3}$ Sv/year for Nicaragua and Honduras. According to the model results for October, Honduras and Guatemala have been experiencing a decrease in moisture contributions from the Caribbean Sea, with trends of $-7\times10^{-3}$ and $-10\times10^{-3}$ Sv/year, respectively. In

contrast, during late autumn Panama shows an increasing trend of $4\times10^{-3}$ Sv/year. The results reveal an opposite response





during the two rainy seasons for the trends in Honduras. This suggests a trend to wetter (drier) for the first (second) part of the rainy season. The results also show a strong drying trend for northern Central America.

The ETPac has become a more active moisture supply over the last three decades. A marked increasing trend up to $10 \times 10^{-3}$

Sv/year for southernmost Central America is observed for summer and autumn (Fig. 8.b). The strongest trends for moisture supply to precipitation were detected for Panama. Because contributions from the ETPac are reportedly on the increase, this result may imply an intensification for the second part of the rainy season in this area. Moreover, because the summer months were identified as seeing the minimum moisture supply (probably associated with the MSD), if the increasing trend in ETPac moisture contributions continues this could imply that mid-summer dry conditions will become milder in future, in

contrast to the findings of Rauscher et al. (2011). The reduction in moisture supply from the Caribbean to Guatemala and Honduras (blue circles in Fig. 8.b) along with the intensification of moisture transport from the ETPac (red and magenta circles in Fig. 8.b) during the second part of the rainy season is highlighted. According to these results, it can be argued that the moisture supply is contributing to the enlargement of the north-south rainfall seesaw in Central America. The detected long-term trend also shows some similarity to the results for future climate based on IPCC models. Future climate

projections suggest the drying of northern Central America, in contrast with wetter conditions for southern Central America under ENSO scenarios (e.g., *Steinhoff et al., 2015*).

Moisture transport from the GoM also exhibits an increasing trend in October. In this case, the strongest increasing trend in moisture supply was found for Guatemala (Fig. 9.c). Moisture contributions from northern South America reveal an overall

negative trend (Fig. 8.d). Based on these results, Panama is the country most affected by the reduction in moisture supply from this source (up to $-6 \times 10^{-3}$ Sv/year). The detected reduction in the moisture supply accords with a decrease in the moisture availability due to the observed intensification of the hydrological cycle over the Amazon, as reported by *Gloor et al (2013)*. From rain gauge data, *Carmona & Poveda (2015)* reported an increase in rainfall in the Colombian Pacific and associated river discharges. They proposed the 'rich-get-richer' mechanism as an explanation for this, in that the increasing

rainfall trends detected here might be accompanied by a reduction in moisture availability for Central American transport. This shows some consistency with the decreasing trend in moisture supply from northern South America (Fig. 8.d)

As previously mentioned, regardless of the fact that positive temperature trends have been detected, the results for precipitation trends lack significance. One of the main constraints in identifying trends in precipitation is the scarcity of

long-term records. To determine whether moisture supply trends can be connected with significant precipitation trends, we used the CHIRPS dataset as a first attempt. CHIRPS precipitation trends were computed, and the significance of the detected trends was based on the rejection of the null hypothesis criteria under a Mann-Kendall test. The results shown in Fig. 9 show the value of the Sen's slope only where significance was proven. Significant precipitation trends for the period 1981-2012, based on CHIRPS data, were detected mostly for the Caribbean slope of Central America. A decrease in precipitation greater



than 6 mm/month per year during the summer months was found, affecting eastern Nicaragua, Honduras and northern Costa Rica (green and blue contours in Fig. 9.a). Positive trends of the same order of magnitude were detected during November for the Caribbean coast of Costa Rica and some regions of Panama (red and pink shading in Fig. 9.b). Regardless of the significance of these trends, the values are in some cases negligible compared to the monthly accumulates for the region (greater than 400 mm). There is thus no conclusive evidence to establish formal links between moisture supply and

precipitation trends. In light of recent records we suggest that large-scale transport processes cannot explain changes in rainfall and that shorter-scale processes such as deep convection and transients must be taken into account in order to explain trends in precipitation. For instance, it is important to know the extent to which extreme events are changing over extended time periods. One other important aspect to consider is how the detected negative trend in moisture supply from the Caribbean to northern Central America is related to ENSO frequency.

**4 Summary**

Based on a dataset of global trajectories, it has been possible to identify moisture sources for Central American precipitation over the long term. As a contribution to an earlier analysis (Durán-Quesada et al., 2010), the present study allowed the generation of a long term climatology of moisture sources for the region. The role of the

Caribbean Sea as the main moisture supply is highlighted, and is in agreement with previous studies. Despite the fact that moisture contributions from the ETPac were confirmed to be relatively small compared to moisture exports from the Caribbean, it was determined that the seasonality of this source is a key driver of the annual cycle of precipitation. The strong link between the local excess of evaporation over precipitation, together with satellite estimates of evapotranspiration, suggest that intense moisture recycling is crucial in enhancing the first

part of the rainy season. However, a detailed quantification of moisture recycling is needed to provide further evidence of the local moisture feedback. Moreover, it is important to determine the role of moisture recycling in the connection between precipitation and vegetation. This will motivate future studies of precipitation-vegetation coupling during the first part of the rainy season in the region, which could provide information on how vegetation cover and land use may have a deep impact on long-term precipitation trends.


The approach used here was found to be very useful for distinguishing the spatial scale of moisture transport, as well as for assessing the oceanic or continental origin of the moisture linked to Central American precipitation. The quantification of the moisture supply from the identified sources supports the explanation of the regional distribution of precipitation as forced by different process scales. It was determined that the first part of the rainy

season is driven at a strongly local scale, while the second part of the rainy season is primarily driven by large-scale processes. The role of the regional low level jets as moisture conveyors is remarkable, as is their influence



on enhanced surface evaporation. From this perspective, we suggest the need to improve our understanding of the impact of regional low level jets in modulating moisture advection within the boundary layer, and the impact of this on heat flux transfer.


The results from this study of the moisture supply from northern South America point to the Magdalena and Orinoco river basins as relevant moisture sources for the summer months. The analysis of OLR suggests a 2-month lag between deep convection over north, north-west and central Amazon, and the moisture supply from northern South America. Whether Amazon convection plays a role in moisture transport through its connection

with the Orinoco and Magdalena basins or by means of another mechanism is not clear. Regardless, the OLR patterns do not appear to provide sufficient evidence of a connecting bridge between Central American weather and climate and Amazon convection related to meridional circulation.

Shown results continuosly remark the importance of the regional LLJs and the interbasin feedback to make the regional precipitation distribution that unique. Being rainfall perhaps the most misrepresented parameter in the

models, to pursue the understanding of the CLLJ and CJ origin is fundamental for model improvement in the area. It is suggested that the CLLJ-CJ bridge should be analyzed from the large scale perspective, this is, evaluate how the ITCZ dynamics can provide information on the interaction of these two structures.

The influence of ENSO in the region must be considered carefully, despite its relatively small size, because

variables such as precipitation are very complex and involve many different scales. From a broad perspective, the sources of moisture show an opposite response to warm and cold ENSO events, and the intensity of this response is sensitive to the location and origin of moisture. Oceanic sources show a larger response to ENSO compared to the continental moisture sourced identified. Positive correlations were detected between the Niño 34 index and moisture transport from the Caribbean (ETPac) during summer (winter) months. It is worth noting that a negative

link was found for the Caribbean Sea transport during the spring months, while a similar response was detected during August for transport from the ETPac. The Increase of summer moisture transport from the GoM to southernmost Central America was detected during warm ENSO events. ENSO forcing of continental moisture transport to Central American precipitation is mostly positive. Local contributions show a strong winter response, while for the case of northern South American transport, this relationship is mostly significant during January and

February (June to August) for Nicaragua and Costa Rica (Costa Rica and Panama). It might be expected that the horizontal precipitation seesaw is reinforced, depending on the ENSO phase.



A long term trend analysis for the moisture supply to Central American countries shows that moisture transport from the Caribbean Sea to Nicaragua and Honduras (Honduras and Guatemala) intensifies (decreases) during March (October) by 7 x $10^{-3}$ Sv/yr. Results for the ETPac moisture source suggest an increase in moisture

transport, mostly to southern Central America, during summer and early autumn. Significant trends of up to 10 x $10^{-3}$ Sv/yr were found for transport from this source to Panama. This result may have important implications in that it can be linked with an intensification of the Chocó Jet, and further changes in the latitudinal position of the ITCZ. It was also determined that moisture transport from the Gulf of Mexico to Guatemala increases during October. It is suggested that this increase may be related to a higher frequency of cold surge activity within the

GoM and a deceleration of the trade winds. It was also determined that moisture transport from northern South America to the region is decreasing, in good agreement with the reported intensification in precipitation over the Amazon basin.

**Acknowledgements**

This research was supported by UCR projects VI805B3600 and VI805B5295 in cooperation with EphysLab

(UVigo). Partial funding was also provided by local grants UCR-VI-805-B0-065, A8-606, B0-130, A9-224, A7-002,and 808-A9-180. Support from student E. Rodríguez (UCR) is acknowledged. Global FLEXPART simulations were computed and processed in the Tsaheva cluster from CIGEFI HPC facilities. The authors thank E. Alfaro (UCR) and H. Hidalgo (UCR) for discussions on the preliminary version of the manuscript as well as O. García (UVigo) for data retrieval support.

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

**Figure 1: Positive (E-P)[-10] seasonal climatology for 1980-2013 (shaded contours) and 925 hPa winds from ERA Interim. Colored areas show the presence of an evaporative source of moisture linked with precipitation over Central America. The high intensity of**
**the Caribbean Sea as a source of moisture is observed all year round while the contributions of the Eastern Tropical Pacific and**





the GoM are strongly seasonally constrained. Wind vectors indicate the boundary between the oceanic sources of moisture identified for Central America and the regional low level jets, CLLJ and CJ.

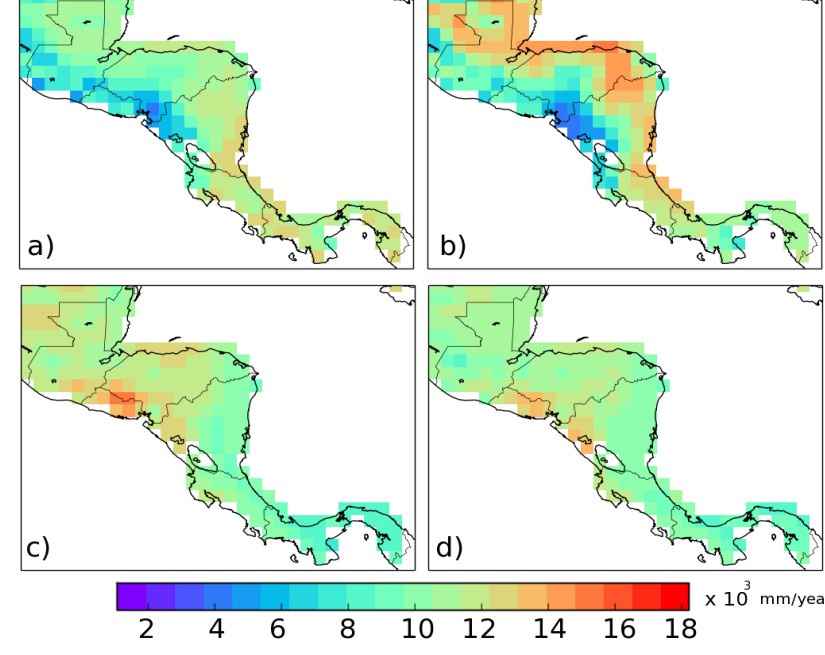

Figure 2: Seasonal average of MODIS-estimated evapotranspiration for 2000-2010. a) DJF, b) MAM, c) JJA and d) SON. Despite the coarse resolution, it can be seen that evapotranspiration presents a strong seasonality. Marked contrasts during the months containing the first peak of the rainy season are observed between the Caribbean and Pacific slopes. The latter is coherent with known higher precipitation and evaporation rates over the Caribbean slope compared to the Pacific influence.





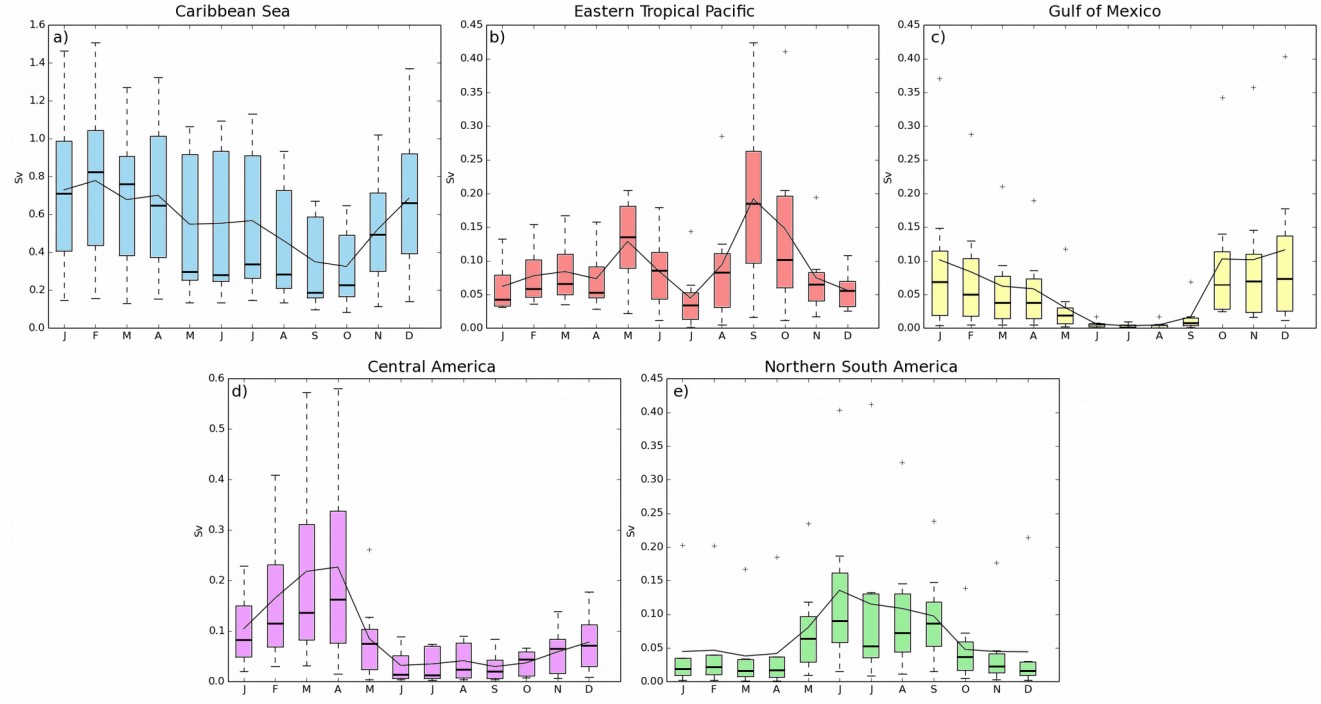

**Figure 3:** **Box plots providing information on the annual cycle of moisture transport from the identified evaporative moisture sources to Central American precipitation, a) Caribbean Sea, b) ETPac, c) GoM, d) Central America and d) Northern South America (Orinoco and Magdalena river basins). The mean is shown as a black line, and the + marks, the narrow black line, and the lower and upper box limits represent outliers, 50, 25 and 75 percentiles respectively. Beyond the determined annual cycle, a general measure of interannual variability is also given by the spread. Note that the vertical scale is different for each panel.**






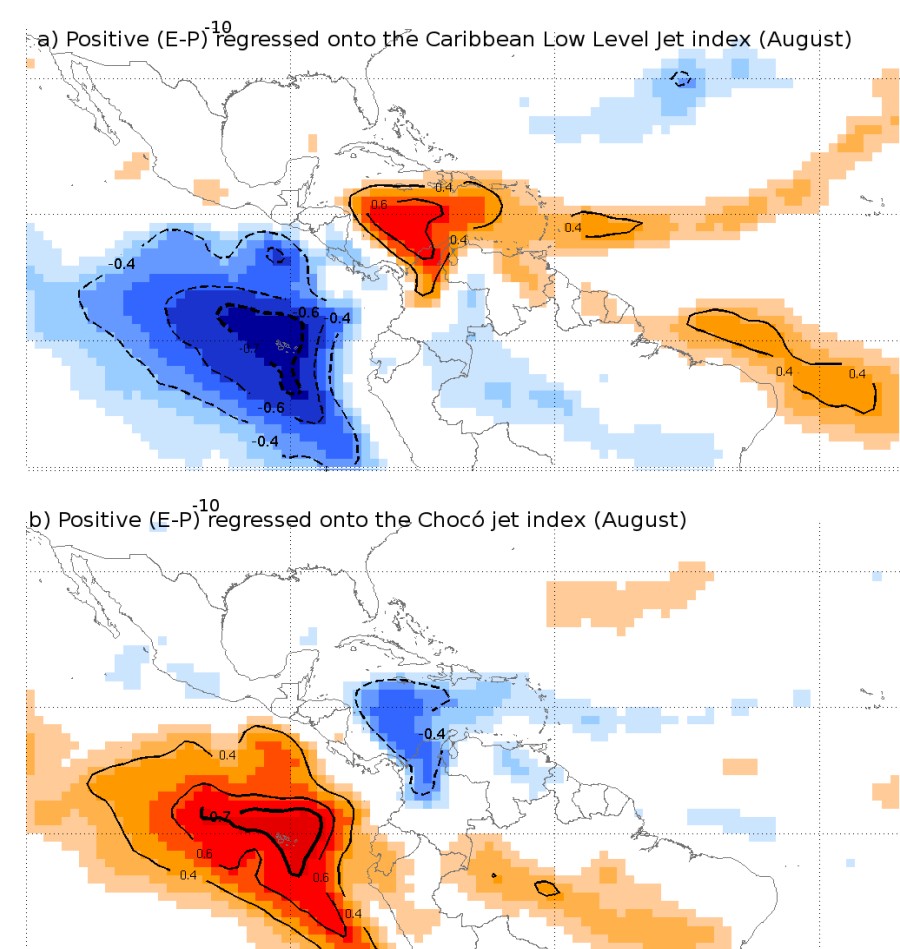



**Figure 4: Positive (E-P)$^{-10}$ regressed against the CLLJ index (upper panel) and the CJ index (lower panel) for August. Only significant values are shaded and contours are indicated for correlations larger than 0.4. Blue shading in the left-hand panel shows a strong negative correlation between the CLLJ and the ETPac sources but a positive correlation with the Caribbean Sea source. Right-hand panel shows the opposite, highlighting a large positive correlation between the activity of the moisture supply from the ETPac source and the CJ index.**







**Figure 5: Spatial correlation of moisture transport from the identified sources and CHIRPS rainfall for the 1981-2013 period.**
**Only correlation coefficients larger than +/- 0.2 and significant at the 90% level are shown. Months for which the correlation**
**patterns were more coherent were chosen. Positive correlation is to be interpret as how the net moisture supply contributes with**
**rainfall whereas negative values require a more careful interpretation. Notice that, for this case, the negative correlation values are**
**more related with the moisture conveyor mechanism that with rainfall itself. Here negative values from 5.a to 5. h suggest that**
**conditions featured by a reduced influence of the moisture mechanism is more likely to favor rainfall in the regions. Fig. 5.i has a**
**different interpretation and relies on the fact that as moisture is evaporated from the continental region the amount of available**
**moisture to contribute with rainfall decreases.**








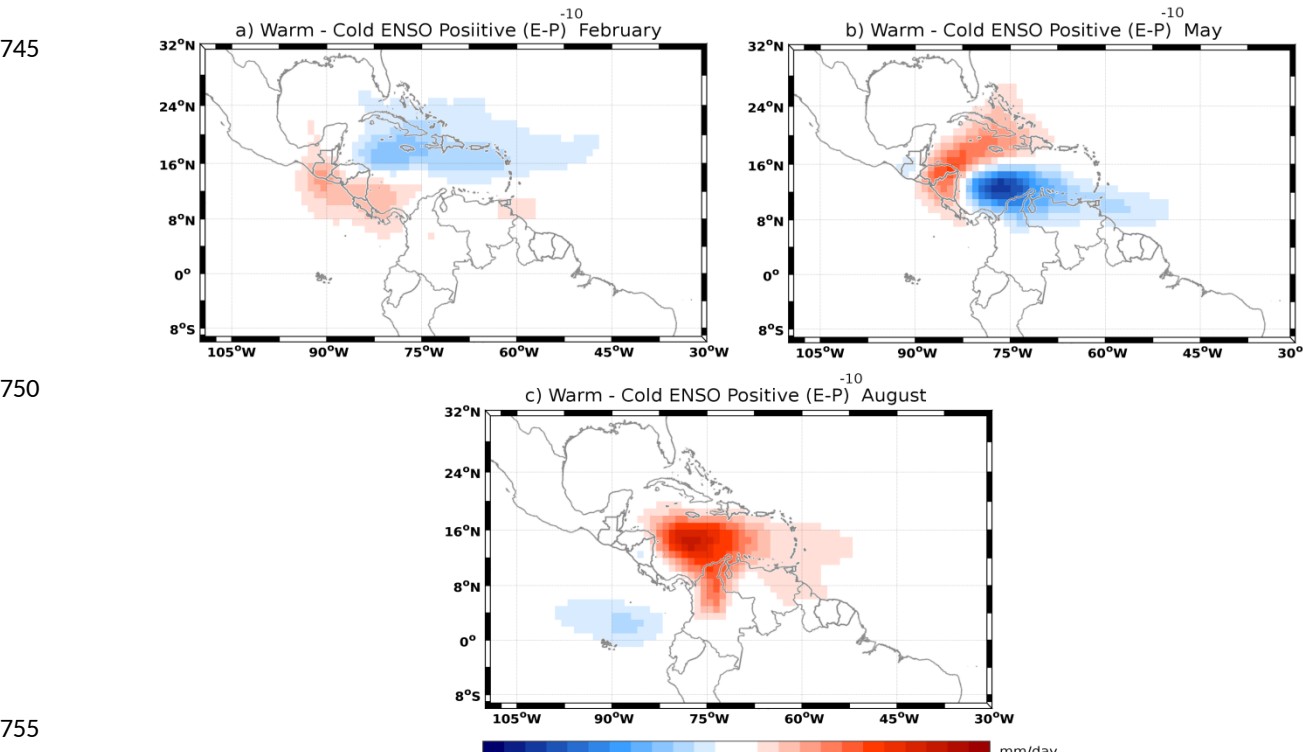


**Figure 6: Positive (E-P)$^{-10}$ composites differences for warm and cold ENSO events based on a +/- 0.8 C SST anomaly for the MEI using the 1981-2010 base line for a) February, b) May and c) August. Blue colours represent a decrease in the evaporative source of moisture for warm ENSO while red colours represent the intensification of the evaporative moisture source under warm ENSO conditions.**





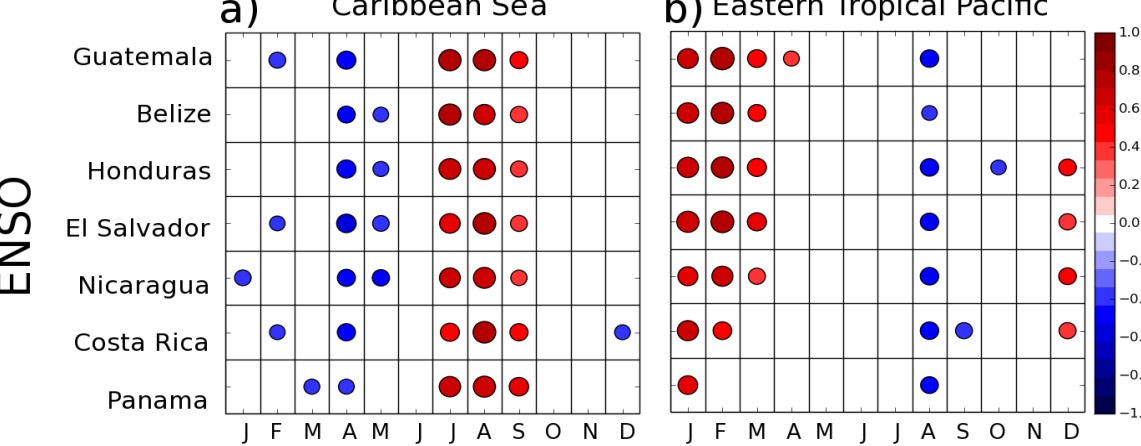

**Figure 7: Correlation coefficients between moisture supply from the a) Caribbean Sea and b) the Eastern Tropical Pacific to precipitation in Central American countries and the MEI index. Only significant correlations are depicted, colour and circle size are proportional to correlation between -1 and 1. Blue (red) indicates negative (positive) correlations.**

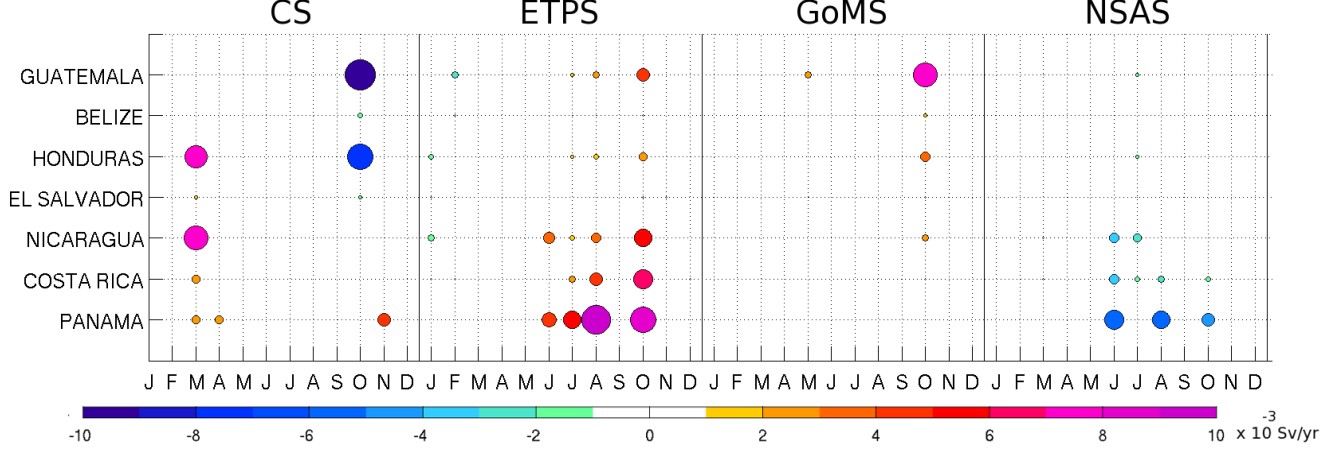



**Figure 8: Long term trends for moisture supply from the evaporative sources a) Caribbean Sea, b) Eastern Tropical Pacific, c) GoM, and d) Northern South America. Note that the evaporative source located over Central America was not considered for this part of the analysis as we have no further details of the observed evapotranspiration trends at this stage in our research. Negative (positive) values represent decreasing (increasing) trends, only those with statistical significance were plotted. Dot size is proportional to trend magnitude.**


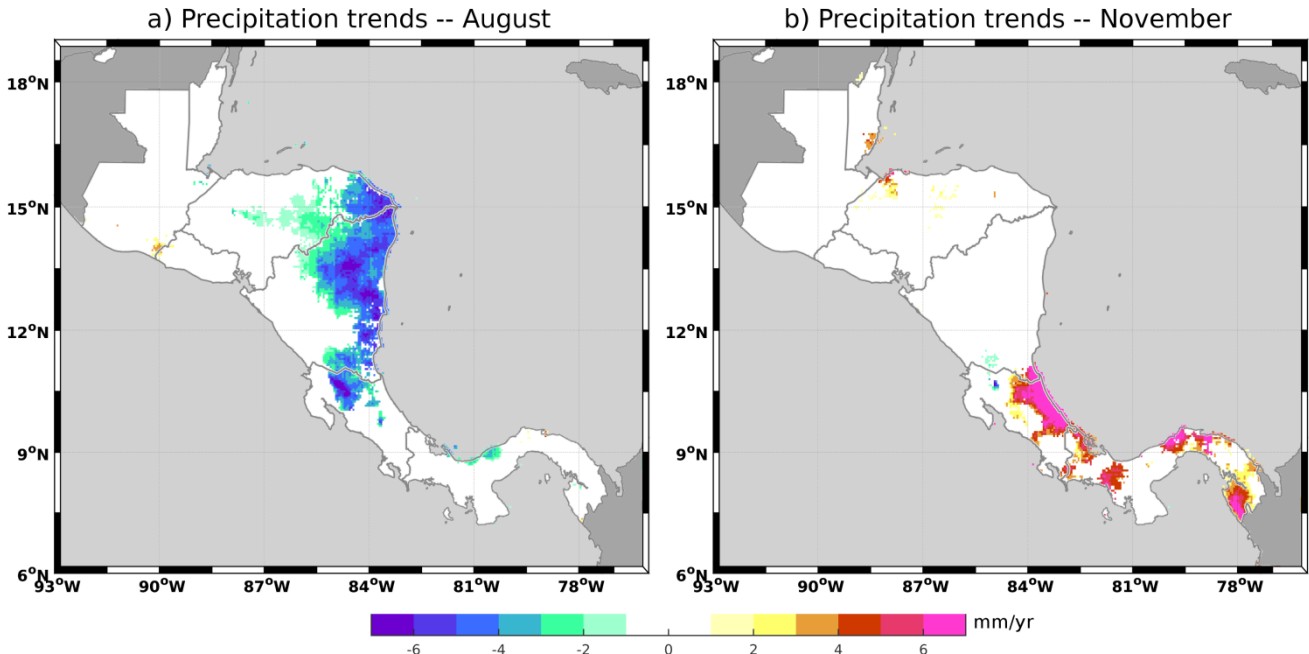

**Figure 9: Long term significant CHIRPS precipitation trends for a) August and b) November. Sen's slope is shown by the shading in units of mm per month every year. Even where significant trends were identified, these trends can be considered very small in comparison to the order of magnitude of monthly climatological values for those regions, which exceed 400 mm/month. The available data is not sufficient to explore the connection between long-term changes in moisture supply and detected rainfall trends.**
