# Peer review of "Role of moisture transport for Central American precipitation"

_Earth System Dynamics, 2016_

## Referee Comment (RC1) · Anonymous Referee #1 · 5 Jan 2017

This paper addresses an important topic regarding the role of moisture transport mechanisms on precipitation over the Central American continent. The paper provides an updated literature review and further analyses on moisture transport over the region by implementing a water vapor tracer model (FLEXPART), providing new elements for our understanding of atmospheric moisture dynamics and variability in Central America and the Intra-American sea (IAS) region. Moreover, the paper aims to address a discussion regarding the role of the low-level jets (Caribbean (CLLJ) and Choco (CJ) jets) on regional precipitation variability, and the further connection between the IAS region, particulary between Northern South America, Central America, and the Caribbean.

The paper is well organized, although I consider necessary to revise the manuscript by a native English speaker, since there are several typos and punctuation errors. Figures are generally clear and provide the necessary information to be understood by a reader.

[Figure]

In my opinion, although the paper provides interesting results and discussion, it needs a further revision in order to be published in Earth System Dynamics.

Specific comments:

1. Page 8 and section 3.3 give some suggestions on how northern South America could contribute to moisture transport toward Central America, with particular emphasis on the role of the CLLJ. Several previous studies are cited. I suggest to include the study by Arias et al. (2015), who used another semi-lagrangian water vapor tracer model, and obtain results suggesting that a weakening of the CLLJ during La Niña years enhance moisture transport toward northern South America, reducing moisture transport to Central America. This is an important coincidence among both studies, since they use different modeling approaches and moisture sources regions.

Arias, P.A., J.A., Martínez, & S.C. Vieira, 2015: Moisture sources to the 2010-2012 anomalous wet season in northern South America. Climate Dynamics, 45(9-10), 2861-2884.

2. Line 179: Moisture transport is quantified using Sv units. Since this is a unit commonly used in oceanography but not in atmospheric sciences, I suggest to give its equivalent in the SI system.

3. Line 239: Several of the conclusions presented in the document are based on OLR analyses, however those are not shown in figures. I consider they sould be included by adding a new figure or modifying current figures.

4. Figure 2: I suggest to include the location of the different national parks mentioned in the text since not all readers may be familiar with the region.

5. Figure 5 and related text: I do not find easy to follow the interpretation of the negative correlations shown in this figure, as explained in figure caption and the main text. I suggest to include further analysis supporting the physical interpretation of these correlations. Also, which statistical test was used for significant correlations? Does the test

account for autocorrelated data? A short explanation of the test should be provided in the methodology section.

6. Figure 6: The figure caption indicates that El Niño and La Niña events were obtained using a threshold of +/- 0.8 °C for the MEI, whereas main text indicates 0.75 °C. This should be clarified.

Technical comments:

The manuscript should be revised by a native English speaker since it contains several typos and punctuation errors. Some comments regarding typos are:

1. In line 29, I suggest to use "natural phenomena-related disasters" or "disasters related to natural phenomena", instead of "natural disasters".

2. In line 34, it should be "regional patterns" instead of "regional patters".

3. In line 39, it should be "represented in numerical simulations." instead of "represented in numerical."

4. Line 115: Sometimes, the authors use "et al" whereas in others they use "et al.". They should use the same sintax.

5. Line 126: It sould be "5 mm/day" instead of "c.5 mm/day".

6. Line 273: It should be "transport from Northern South America accounts..." instead of "transport from Norther South America iaccounts..."

7. Lines 493-495: The reference has a different format.

---

## Referee Comment (RC2) · Anonymous Referee #2 · 8 Jan 2017

The manuscript discusses the climatological behavior of large-scale and regional moisture sources for Central America. The results are clearly explained and are of great relevance to the scientific community in Central America and other regions of the world. I recommend to accept the manuscript after a few minor changes have been made.

General Comments

Due to its importance, and because it is also mentioned at the Introduction, the authors should include a few comments about how they think their results could be used by the forecast community in the near future. Their results are indeed very important for this community and the fact that there is good predictability for certain low-level jets (LLJs) in the region is something other people could take advantage of. Forecast conditioned on the behavior of LLJs and moisture availability from the sources here identified could

be a promising way forward.

The authors should emphasize the period used to compute the trends. Section 3.4 starts with a reference to climate change, and then the trends referenced in the analysis could be confused by some readers to long-term (climate change) trends. Related to that comment, I think it is important to clarify somewhere at the beginning of Section 3.4 that climate change is just a part of the whole signal observed, as natural climate variability has also a role in the analysis performed by the authors.

Specific Comments

L50-51: Please explain what do you mean by "highly active stratified precipitation".

L53-54: The authors can also cite the work of Moron et al. (2016). Moron, V., Gouirand, I. & Taylor, M. Clim Dyn (2016) 47: 601. doi:10.1007/s00382-015-2858-9

L104: Is the CJ index computed also using zonal wind for that region? Or meridional?

L152: Capital "t" for "The transport of moisture..."

L224: The Orinoco river basin is present in both Colombia and Venezuela. I suggest you indicate where the Mapire river mouth is, just to quickly provide a geographical reference to the reader.

L338: As indicated before, I suggest you mention as soon as possible in this section that the trends correspond to a relatively short period of time, so there is no confusion with long-term trends associated with climate change. Also, I recommend you mention that these trends are due to the contribution of climate change AND natural climate variability at different timescales. Future research could analyze the physical mechanisms behind these trends.

L373-374: Yes, but that does not mean that climate change is the only or even the main cause.

L431-432: The authors can cite the work of Mapes et al. (2003) and Munoz et al.

(2016) as concrete examples of how a LLJ modulates moisture availability and promotes (deep) convection.

Mapes et al. Diurnal Patterns of Rainfall in Northwestern South America. Part I: Observations and Context. Mon. Wea. Rev. 131, 799-812, 2003.

Munoz et al. (2016). http://dx.doi.org/10.1016/j.atmosres.2015.12.018

L451: The increase. . .

Figure 4 (caption): I think the authors mean "lower panel" rather than "right hand" panel.

---

## Referee Comment (RC3) · Anonymous Referee #3 · 9 Jan 2017

The paper is a very good and sound discussion on moisture transport over Central America, sources of precipitation, relation to ENSO and a brief but sound description of trends It is well written and uses sound data and methods. Figures and clear and well laid out. One consideration: when country-wide correlations are described, how is data integrated into a regional aggregate? How are internal variations evaluated? A few typos and minor considerations Line 34: patters ==> patterns L39: in numerical ==> ??? missing word? L75: As follows, ==> as follows: L125: mm/day. . It is notable ==> mm/day. It is noticeable L160: In Mexico . ==> In Mexico. L160: (E-P)-10: please, define all acronyms, even if they are obvious to most readers L266: Central American coas ==> Central American Coast L274: conveyor , ==> conveyor, L280: using the MEI ==>, again, please, define all acronyms L296: complimentary ==> complementary L391: significance was proven ==> was reached at xx level ...

---

## Author Comment (AC1) · 24 Jan 2017

Referee 1.

Thank you for your valuable time, please find the response to your comments:

Specific comments

1. Paper Arias, P.A., J.A., Martínez, & S.C. Vieira, 2015: Moisture sources to the 2010-2012 anomalous wet season in northern South America. Climate Dynamics, 45(9-10), 2861- 2884 was carefully read and the correspondent cite was incorporated with additional comments on the relationship between the cited paper and the present study.

2. The authors were previously advised to used Sv units as it is the unit used for

transport in this context, however we included a brief description on the Sv to SI units conversion to clarify for a broader audience.

3. A new panel showing the annual cycle of the computed OLR indices was included in figure 3 as 3.f and the correspondent figure referencing was included in the discussion.

4. The location of the parks is now included.

5. The interpretation was re-written to better explain the results. We used a regular Mann-Kendall test and also a modified Mann-Kendall test for autocorrelated data. This is now clarified in the methods section.

6. There was an error in the caption and is now corrected.

Technical comments

The discussion version of the manuscript was already proofread, anyway after minor modifications were incorporated following the suggestions of the reviewers, the document was sent for proofreading. Format of the references was corrected.

---

## Author Comment (AC2) · 24 Jan 2017

Thank you for your time. Results were computed for "target regions" defined as a) the polygons following the continental boundaries for each country and b) the polygons following the regional boundaries. Analysis of the results included to check consistency between the integration of the countries results and the results for the complete region. Results were almost the same, variation was about 0.17% which is attributed to the computational handling of the data integration (sum). Internal variations were evaluated using noise-signal detection analysis, the results indicated there were no statistically significant internal variations. Typos were corrected.

---

## Author Comment (AC3) · 24 Jan 2017

Thank you for your time and contributions. Trends computation period is now indicated as well as a very brief explanation on the climate change and natural variability differences.

Regarding the specific comments: 1- There was a typo, we meant "stratiform" and not "stratified". "Highly active stratiform precipitation" means that the region is under a strong influence of large cloud systems that account for a fraction of the observed precipitation.

2- Citation of the suggested work by Moron et al. (2016) is included and a short discussion of the relevance of their work to the analysis proposed can be found.

[Figure]

3- In the methods section we explain how the CJ was computed.

4- Typo corrected.

5- We now use the Mapire river mouth as suggested.

6- We provide a extended explanation of the brief comment on climate change – climate variability differences mentioned in the methods section.

7- Indeed, we explain now this better to avoid confusion as one of the main results we want to highlight is that the effect of interannual variability on the moisture supply is more significant that the trends detected and therefore a focus on interannual variability would be great information input for planning and decision making processes.

8- Works from Mapes et al. (2003) and Munoz et al (2016) are now incorporated in the discussion and cited.

9- Figure caption modified.